# Rare transmission of commensal and pathogenic bacteria in the gut microbiome of hospitalized adults

Benjamin A. Siranosian [1], Erin F. Brooks [2], Tessa Andermann [3], Andrew R. Rezvani[4], Niaz Banaei[5,6,7], Hua Tang [1] & Ami S. Bhatt [1,2,4✉]

Bacterial bloodstream infections are a major cause of morbidity and mortality among patients undergoing hematopoietic cell transplantation (HCT). Although previous research has demonstrated that pathogens may translocate from the gut microbiome into the bloodstream to cause infections, the mechanisms by which HCT patients acquire pathogens in their microbiome have not yet been described. Here, we use linked-read and short-read metagenomic sequencing to analyze 401 stool samples collected from 149 adults undergoing HCT and hospitalized in the same unit over three years, many of whom were roommates. We use metagenomic assembly and strain-specific comparison methods to search for high-identity bacterial strains, which may indicate transmission between the gut microbiomes of patients. Overall, the microbiomes of patients who share time and space in the hospital do not converge in taxonomic composition. However, we do observe six pairs of patients who harbor identical or nearly identical strains of the pathogen *Enterococcus faecium*, or the gut commensals *Akkermansia muciniphila* and *Hungatella hathewayi*. These shared strains may result from direct transmission between patients who shared a room and bathroom, acquisition from a common hospital source, or transmission from an unsampled intermediate. We also identify multiple patients with identical strains of species commonly found in commercial probiotics, including *Lactobacillus rhamnosus* and *Streptococcus thermophilus*. In summary, our findings indicate that sharing of identical pathogens between the gut microbiomes of multiple patients is a rare phenomenon. Furthermore, the observed potential transmission of commensal, immunomodulatory microbes suggests that exposure to other humans may contribute to microbiome reassembly post-HCT.

[1] Department of Genetics, Stanford University, Stanford, CA, USA. [2] Department of Medicine, Division of Hematology, Stanford University, Stanford, CA, USA. [3] Division of Infectious Diseases, University of North Carolina at Chapel Hill, Chapel Hill, NC, USA. [4] Department of Medicine, Division of Blood and Marrow Transplantation and Cellular Therapy, Stanford University School of Medicine, Stanford, CA, USA. [5] Department of Medicine, Division of Infectious Diseases and Geographic Medicine, Stanford University, Stanford, CA, USA. [6] Clinical Microbiology Laboratory, Stanford University Medical Center, Stanford, CA, USA. [7] Department of Pathology, Stanford University, Stanford, CA, USA. ✉email: asbhatt@stanford.edu

Patients undergoing hematopoietic cell transplantation (HCT), a potentially curative treatment for a range of hematologic malignancies and disorders, are at increased risk for bloodstream infections (BSIs) and associated morbidity and mortality[1]. While the bacterial pathogens that cause BSIs in HCT patients are well understood, their routes of transmission are often unclear. Determining these transmission pathways involves identifying two critical elements: the source of the infection, i.e., how the pathogen was introduced into the patient's bloodstream, and the origins of the particular pathogen causing the BSI.

The most common pathways by which bacterial pathogens can be introduced into an HCT patient's bloodstream include contaminated central intravenous lines and translocation of intestinal microbiota across a damaged epithelium[2]. Indeed, research from our group and others has shown that strains of bacteria isolated from the blood of HCT patients with BSIs may be indistinguishable from the strains in the intestinal microbiota of these patients prior to infection[3–5]. In addition, HCT patients with a microbiome dominated by a single bacterial taxon, such as Enterococcus or Streptococcus, are at increased risk for not only BSI[6,7], but also graft-versus-host disease[8,9] and death[10–13].

Identifying the source of the BSI is only the first step. To fully understand the transmission pathways of bacterial pathogens in hospital settings, it is also essential to determine the origin of the pathogen that caused the BSI. For gut-based pathogens, there are three possibilities. First, they may exist in the HCT patient's microbiome upon admission to the hospital. Second, hospital environments and equipment may serve as unintentional reservoirs of pathogens, thereby infecting multiple patients through exposure[14]. Lastly, a pathogen could originate from the microbiome of other patients, healthcare workers or hospital visitors and be transmitted via person-to-person contact. In cases where traditional epidemiological links cannot be found, this patient–patient transmission of gut microbes may be the "missing link" that explains the persistence of BSIs in hospital environments[15].

Transmission of gut bacteria and phages between individuals is known to occur in specific cases, such as from mothers to developing infants[16–18]. By contrast, adults have a microbiome that is relatively resistant to colonization with new organisms even after perturbation by antibiotics[19–21]. While adults living in the same household or in close-knit communities may have more similar microbes than those outside the group[22], to our knowledge, direct transmission of gut microbes between adults has not been observed with high-resolution metagenomic methods, with the notable exception of fecal microbiota transplantation[23–25], a drastic reshaping of the gut microbiota often used in response to Clostridioides difficile infection. Transmission of gut microbiota is thought to occur by a fecal–oral route, which could happen in the hospital environment by exposure to contaminated surfaces or equipment, sharing a room or bathroom, contaminated hands of healthcare workers, or other sources. The perturbed microbiomes of HCT patients, often lacking key species to provide colonization resistance, may be primed to acquire new species from these sources.

Previous studies of the microbiome in HCT patients have often used 16S rRNA sequencing[10,26–28], which is sufficient for taxonomic classification but may not be sensitive enough to differentiate strains with similar genomes. By contrast, short-read shotgun metagenomic sequencing can capture information from all bacterial, archaeal, eukaryotic, and phage DNA in a stool sample. While short-read sequencing data is accurate on a per-base level, it is often insufficient to assemble complete bacterial genomes due to the presence of repeated genetic elements. Linked-read sequencing captures additional long-range

information by introducing molecular barcodes in the library preparation step. This technology allows for significant increases in assembly contiguity[29,30] while retaining high per-base accuracy. Both of these technologies also capture information about strain diversity, genetic variation within the population of a species[31,32], which is critical for measuring transmission between microbiomes.

Here, we use a collection of short-read and linked-read metagenomic sequencing datasets from 401 stool samples to analyze bacterial transmission between HCT patient microbiomes at a single, high-volume hospital. We apply strain-resolved comparison methods to show that transmission of bacteria between adults hospitalized in the same unit at the same time is likely a rare event, usually occurring when recipients have extremely perturbed microbiomes, such as after exposure to broad-spectrum antibiotics. Bacterial strains shared between individuals include both pathogenic and commensal organisms, demonstrating that transmission may depend more on niche availability than pathogenicity or antibiotic resistance capacity. We find that pathogens colonizing HCT patient microbiomes are present in the first sample in a time course roughly 60–70% of the time in our cohort. This suggests that in most cases, prior colonization, rather than direct transmission from other patients or the hospital environment, is responsible for pathogenic organisms in the gut microbiomes of this patient population. Even though patients were frequently placed into double occupancy hospital rooms with a shared bathroom, we observe relatively few putative transmission events. This implies that sharing a room with another patient may not place a patient recovering from HCT at a greatly increased risk of acquiring pathogens in their gut microbiome.

## Results

**Sample characteristics and patient geography.** We collected weekly stool samples (see the "Methods" section) from adult patients undergoing HCT at Stanford University Medical Center from 2015 to 2019. At the time of the study, our biobank contained over 2000 stool samples from over 900 patients. Samples from October 2015 to November 2018 were considered for this study. Relevant patient health, medication, demographic, hospital admission, and room occupancy data were extracted from electronic health records (Table 1, Supplementary Data 1). All patients stayed in a single ward of the hospital during treatment, which contained 14 single-occupancy and four double-occupancy rooms, the latter of which included shared bathrooms (Fig. S1a). Patients spent a median of 18 days on the ward and were frequently moved between rooms: 42% of patients spent at least one day in three or more rooms during treatment (Table 2, Fig. S1c). 73% of patients shared a room with a roommate for ≥24 h. Over the course of their hospital stays, many patients had several roommates, though never more than one at a time (Fig. S1d). Patients with multi-drug resistant Gram-negative infections were always placed into single rooms with contact precautions. We use the term "hospital overlap" to refer to patients who are in the hospital at the same time. "Roommate overlap" refers to patients who are in the same room at the same time and is a subset of hospital overlap. When not specified otherwise, a minimum time of 24 h was considered for both hospital and roommate overlap.

To understand how roommate overlap may influence transmission of gut microbes, we created a network from patient–roommate interactions lasting at least 24 h (Fig. S1b). 535 patients (77% of patients with at least one roommate, 56% of all patients) fell into the largest connected component of the network. Although the largest component was not densely connected (mean degree 2.2 ± 1.6 standard deviation (SD)), it

**Table 1 Aggregated characteristics of patients with samples investigated in this study.**

| Attribute | n | % |
|---|---|---|
| Total sequenced patients | 149 | 100 |
| *Age* | | |
| ≤30 | 9 | 6% |
| 31–40 | 17 | 11% |
| 41–50 | 24 | 16% |
| 51–60 | 38 | 26% |
| 61–70 | 55 | 37% |
| ≥71 | 6 | 4% |
| *Sex* | | |
| Sex Male | 87 | 58% |
| *Diagnosis* | | |
| ALL: Acute lymphocytic leukemia | 21 | 14% |
| AML: Acute myelogenous leukemia | 42 | 28% |
| CML: Chronic myeloid leukemia | 6 | 4% |
| HL: Hodgkin lymphoma | 4 | 3% |
| MDS: Myelodysplastic syndrome | 42 | 28% |
| NHL: Non-Hodgkin lymphoma | 23 | 15% |
| OTHER: Other malignancy | 11 | 7% |
| *Graft* | | |
| Allogeneic | 132 | 89% |
| *GVHD* | | |
| Accute GVHD yes | 88 | 59% |
| Chronic GVHD yes | 29 | 19% |
| *Bloodstream infection (BSI) Genera, within 0–180 days after HCT* | | |
| Any BSI | 54 | 36 |
| *Bacillus* | 1 | 1 |
| *Enterobacter* | 2 | 1 |
| *Enterococcus* | 7 | 5 |
| *Escherichia* | 8 | 5 |
| *Gemella* | 1 | 1 |
| *Klebsiella* | 6 | 4 |
| *Pseudomonas* | 2 | 1 |
| *Rothia* | 4 | 3 |
| *Staphylococcus* | 14 | 9 |
| *Streptococcus* | 9 | 6 |

**Table 2 Aggregated statistics of temporal geographic data for all patients on the ward during the study period.**

| | All patients | At least one sequenced sample |
|---|---|---|
| Number of patients ≥24 h on the HCT ward | 923 | 149 |
| *Days spent as an inpatient on the HCT ward* | | |
| Mean | 21.9 | 37.6 |
| Median | 18 | 30.8 |
| SD | 18.2 | 21.8 |
| Min | 1 | 8.7 |
| Max | 175.4 | 137.7 |
| *Number of rooms a patient occupied (minimum time 24 h)* | | |
| Mean | 2.6 | 3.5 |
| Median | 2 | 3 |
| SD | 1.3 | 1.7 |
| Min | 1 | 1 |
| Max | 9 | 9 |
| *Number of unique patients a given patient overlapped within the hospital ward (minimum time 24 h)* | | |
| Mean | 55.9 | 80.8 |
| Median | 48 | 72 |
| SD | 30.7 | 38.6 |
| Min | 8 | 25 |
| Max | 246 | 198 |
| *Number of unique patients a given patient overlapped with as roommates (minimum time 24 h)* | | |
| Mean | 1.5 | 2.5 |
| Median | 1 | 2 |
| SD | 1.5 | 2.4 |
| Min | 0 | 0 |
| Max | 13 | 13 |

*SD* standard deviation.

links together patients over 3 years and may represent a risk for infection transmission. We used the network to select samples for further analysis with metagenomic sequencing, as described in the "Methods" section.

**Metagenomic sequencing, assembly and binning**. For an overview of the steps used in the generation and processing of sequence data (see Fig. 1a). 328 stool samples from 94 HCT patients were subjected to short-read metagenomic sequencing as part of previous projects (for references and SRA IDs of these samples, see Supplementary Data 2). 96 additional samples from 62 patients were selected for linked-read sequencing to span periods of roommate overlap between patients. Samples were subjected to bead beating-based DNA extraction and bead-based DNA size selection for fragments ≥2 kb (see the "Methods" section). We prepared linked-read sequencing libraries with the 10× Genomics Chromium platform from 89 samples with sufficient DNA concentration. Samples were sequenced to a median of 116 million ($M$) (±37M SD) read pairs on an Illumina HiSeq4000. In total, 401 stool samples from 149 patients were sequenced (Table 3), with a median of 2 and maximum of 13 samples per patient (Fig. 1b).

We processed all existing short-read data and newly generated linked-read data by first trimming and then removing low-quality reads, PCR duplicates (short-read data only) and reads that aligned against the human genome (see the "Methods" section).

**Table 3 Aggregated statistics of sequencing datasets and metagenome-assembled genomes (MAGs) generated in this study.**

| Attribute | n | | |
|---|---|---|---|
| Total stool samples sequenced | 401 | | |
| Total sequencing datasets | 405 | | |
| Short-read (SR) | 312 | | |
| Linked-read (LR) | 93 | | |
| Sequenced with SR and LR | 4 | | |
| **Samples sequenced per patient** | **Median** | **Range** | **SD** |
| | 2 | 1–13 | 2.4 |
| **Reads after processing (M)** | **Median** | **Range** | **SD** |
| SR | 7.6 | 0.01–28.8 | 4.4 |
| LR | 104 | 0.9–323.6 | 40 |
| **Assembly N50 (kb)** | **Median** | **Range** | **SD** |
| SR | 17.2 | 0.7–163.6 | 24.8 |
| LR | 147.6 | 9.5–956.3 | 168.5 |
| **Binned genomes** | **n** | **%** | |
| SR | 2859 | | |
| High quality | 103 | 4 | |
| Medium quality | 2124 | 74 | |
| Low quality | 632 | 22 | |
| LR | 1900 | | |
| High quality | 518 | 27 | |
| Medium quality | 950 | 50 | |
| Low quality | 432 | 23 | |

*SD* standard deviation; *M* million; *kb* kilobase pairs.

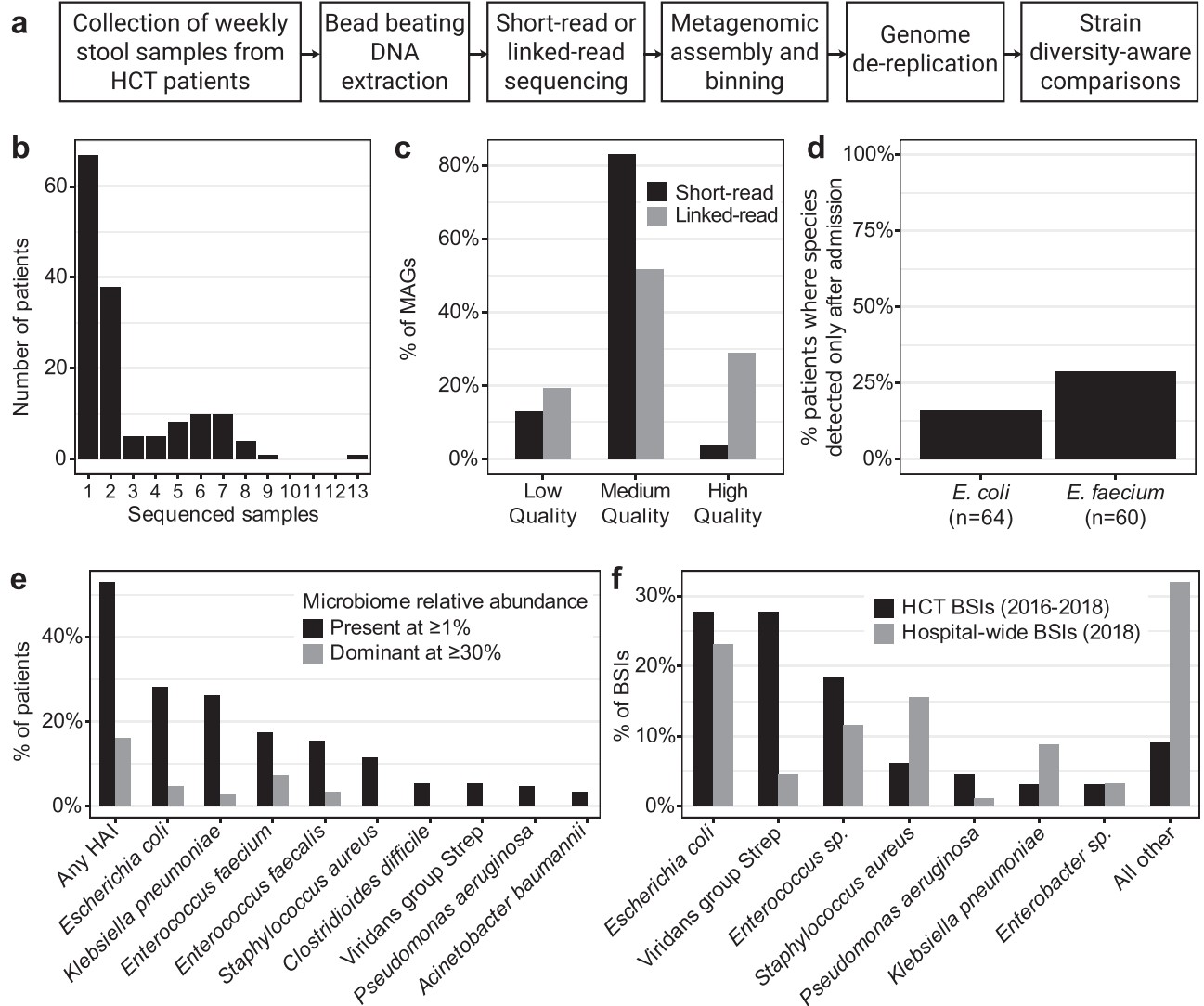

**Fig. 1 Overview of the methods, data generated, and clinical features of this sample set. a** Overview of the experimental and computational workflow used to generate sequencing datasets, bin metagenome-assembled genomes (MAGs), and compare strains between patients. **b** Number of stool samples sequenced per patient. **c** Percentage of MAGs meeting each quality level[40], stratified by sequencing method. **d** Of patients who have the given organism detected (≥50% coverage breadth) in a time course sample, percentage of patients where the organism was below the detection threshold (<50% coverage breadth) in the first sample. **e** Percentage of patients with at least one sample positive with (≥1% relative abundance) or dominated by (≥30% relative abundance) hospital-acquired infection (HAI) organisms, as identified by Kraken2- and Bracken-based classification. **f** Percentage of bloodstream infections (BSIs) identified with each organism or group in HCT patients and hospital-wide. Source data for this figure are provided in Supplementary data 2–4 and Supplementary Table 1.

After quality control, newly sequenced linked-read samples had a median 104 M (±40 M SD) read pairs, while short-read data had a median 7.6 M (±4.4 M SD) read pairs. Metagenomic assembly was conducted using metaSPAdes[33] for short-read data, and MEGAHIT[34] followed by Athena[29] for linked-read data. Short-read assemblies had a median N50 of 17.2 ± 24.8 kb, while linked-read assemblies had a median N50 of 147.6 ± 165.8 kb. We binned metagenome-assembled genomes (MAGs) using Metabat2[35], Maxbin[36] and CONCOCT[37] and aggregated across results from each tool using DASTool[38]. MAG completeness and contamination was evaluated using CheckM[39] and MAG quality was determined by previously established standards[40]. The vast majority of short-read and linked-read MAGs were at least medium quality, and 27% of linked-read MAGs contained the 5S, 16S, and 23S rRNA genes and at least 18 tRNAs to be considered high-quality (Fig. 1c, Supplementary Data 3). Linked-read MAGs had higher quality than the 4644 species-level genomes in the

Unified Human Gastrointestinal Genome collection[41], where 573 genomes (12.3%) are high-quality, and only 38 (6.6%) of those came from metagenomes rather than isolates. Sequencing dataset type (short-read vs. linked-read) was not significantly associated with MAG length (student's t-test, $p > 0.9$); the increase in quality was mainly due to the inclusion of ribosomal and transfer RNA genes in the linked-read MAGs, which often do not assemble well with short-read sequencing data alone. To understand the diversity of strains present in the microbiomes of our patients, we clustered all medium-quality and high-quality MAGs at 95% and 99% identity thresholds (roughly "species" and "strain" level, see the "Methods" section) using dRep[42], yielding 1615 unique genomes representative of the microbial diversity in this sample set.

**Classification of abundant healthcare-associated infection organisms**. We performed taxonomic classification of sequencing

reads with Kraken2[43] and abundance estimation with Bracken[44] using a custom database of bacterial, fungal, archaeal and viral genomes in NCBI Genbank (see the "Methods" section) (Supplementary Data 4 and 5). A median of $33 \pm 15\%$ SD reads were classified to the species level with Kraken2 ($72 \pm 15\%$ SD at the genus level), which was improved to $96 \pm 7\%$ SD using Bracken ($97 \pm 8\%$ SD at the genus level). Organisms that cause healthcare-associated infections (HAI) were identified from the CDC list of pathogens[45]. Here, we report organisms as present if they achieve 1% relative abundance, but acknowledge that many microbes typically exist at lower concentrations, which may be more difficult to detect with metagenomic sequencing.

Many HAI organisms were prevalent in the microbiomes of the studied HCT patients. 152 samples (38%) from 79 patients (53%) had at least one HAI organism identified at 1% relative abundance or above (Fig. 1e). *Escherichia coli* was the most common HAI organism (present at ≥1% in 42/149 patients, 28.2%; ≥0.1% = 80/149, 53.7%), followed by *Klebsiella pneumoniae* (39/149 patients, 26.2%; ≥0.1% = 70/149, 47%) and *Enterococcus faecium* (26/149 patients, 17.4%; ≥0.1% = 73/149, 49%). Rates of colonization with HAI organisms were much higher than in stool samples from healthy individuals in the Human Microbiome Project[46] where *E. coli* reaches 1% relative abundance in 2.1% of samples, and *K. pneumoniae* and *E. faecium* are never found at >1% (present at ≥0.1% in 18.4%, 1.4%, and 0.7% of samples, respectively).

HCT patient microbiomes can become dominated by HAI organisms, often as a result of antibiotic usage. 24 patients (16%) have at least one sample with a dominant HAI organism (≥30% relative abundance), which may place them at increased risk for BSIs[6]. BSI in this cohort of HCT patients is most frequently caused by *E. coli*, viridans group Streptococci and *E. faecium*; these organisms less frequently cause BSI among the entire inpatient population at our hospital (Fig. 1f, Supplementary Table 1). We focused further analysis on *E. coli* and *E. faecium*, as these species are both frequently detected in stool and frequently cause BSIs. While viridans group Streptococci frequently cause BSIs in HCT patients, these species are typically more prevalent and abundant in the oral cavity[2,47] compared to the gut microbiome (individual species in the group only reach 1% relative abundance in 8/149 patients, 5%).

**Detection of *E. coli* and *E. faecium* becomes more common during a patient's hospital stay.** We investigated the detection of *E. coli* and *E. faecium* in patients with time course samples (82/149 patients, 55%) as a proxy for understanding if these organisms were acquired or became more abundant during the observed hospital stay. Of the 1615 de-replicated MAGs, nine were identified as *E. coli* and five were identified as *E. faecium*. We mapped sequencing reads from all samples to these MAGs and evaluated the maximum coverage breadth, the fraction of the reference genome covered with at least one sequencing read. We used a breadth cutoff of 50% to determine "detection" or "absence" in a sample. This threshold is likely specific (it is difficult to achieve 50% breadth by read mis-mapping or homology with a different organism) but not extremely sensitive (it will likely miss very lowly abundant organisms that are truly present).

64/82 (78%) patients with time-course samples have *E. coli* present at 50% breadth in at least one sample. Of these, 10/64 (16%) have *E. coli* below 50% breadth in the first sample. 60/82 (73%) patients with time-course samples have *E. faecium* present at 50% breadth in at least one sample. Of these, 17/60 (28%) have *E. faecium* below 50% breadth in the first sample (Fig. 1d). In these patients, *E. coli* or *E. faecium* may have been newly acquired into the gut microbiome during the hospital stay. Alternatively,

the organism could have also been present at low abundance and below our limit of detection in the first sample. It is possible that antibiotic use in the weeks after HCT kills off many of natural microbiome colonizers, allowing antibiotic-resistant HAI organisms to increase in relative abundance above our limit of detection. Overall, the relative abundance of *E. coli* and *E. faecium* was not significantly different in first samples compared to later samples (Wilcoxon rank-sum test). While more patients have newly detectable *E. faecium* than *E. coli*, the difference was not statistically significant ($p = 0.09$, binomial likelihood ratio test).

**Antibiotic use and its effect on HCT patient microbiomes.** HCT patients are frequently prescribed antibiotic, antiviral and antifungal drugs, especially in the days immediately after transplant. These drugs can have a significant impact on the microbial populations in the gut and contribute to the loss of microbial diversity frequently observed following HCT[6]. Antibiotic use likely impacts the dynamics of bacterial transmission in this patient population, as both natural colonizers, which may provide resistance to newly invading species, and potentially transmitted species can be killed by the drugs. We gathered electronic health record data to understand the characteristics of antibiotic prescription in our patient cohort and its potential impact on the gut microbiome composition (Supplementary Data 6).

Patients were prescribed a median of five different antibiotics (range 1–10) and had a median of 90 cumulative antibiotic-days (range 14–416). The most common antibiotics prescribed were ciprofloxacin (98% of patients prescribed for at least one day), intravenous (IV) vancomycin (80%), cefepime (66%), and piperacillin-tazobactam (57%). Prescription of most antibiotics peaked in the 14 days following HCT, while administration of antifungal drugs like posaconazole and antiviral drugs like ganciclovir were higher up to 50 days following HCT (Fig. 2a). We found that antibiotic usage in the prior seven days before a stool sample was collected was strongly negatively associated with bacterial diversity (linear regression on cumulative antibiotic-days and Shannon diversity, $R^2 = 0.18, 0.14$, $p = 8.9e{-}11, 2.6e{-}8$ for species and genus level, respectively). Samples with a single species dominant at 30% relative abundance or higher also typically came from patients who had higher antibiotic usage in the past seven days prior to that sample ($p = 0.001$, Wilcoxon rank-sum Test) (Fig. 2b–d).

**Patients who share time and space in the hospital do not converge in microbiome composition or frequently share strains.** To understand the overall impact of hospital and roommate overlap on patient microbiomes, we first studied the taxonomic (Bray–Curtis) similarity between samples from different patients. For each sample pair, we calculated the maximum time the two patients had overlapped in the hospital or as roommates, determined by the earlier sample time (Supplementary Data 7 and 8). There was not a significant linear association between time of overlap and taxonomic similarity (linear regression, $p = 0.42$ for roommate, $p = 0.068$ for hospital, Fig. 2e, f), indicating that the broad taxonomic composition of patient microbiomes may not converge under hospital and roommate overlap. However, there may be isolated strains that are shared between patients. We used the strain diversity-aware, SNP-based method inStrain[48] to conduct a sensitive analysis of bacterial strains shared between patients. InStrain compares alignments of short reads from multiple samples to the same reference genome and reports two metrics: Consensus ANI (conANI) and PopulationANI (popANI). ConANI counts a SNP when two samples differ in the consensus allele at a position in the reference genome, similar to many conventional SNP calling methods. PopANI

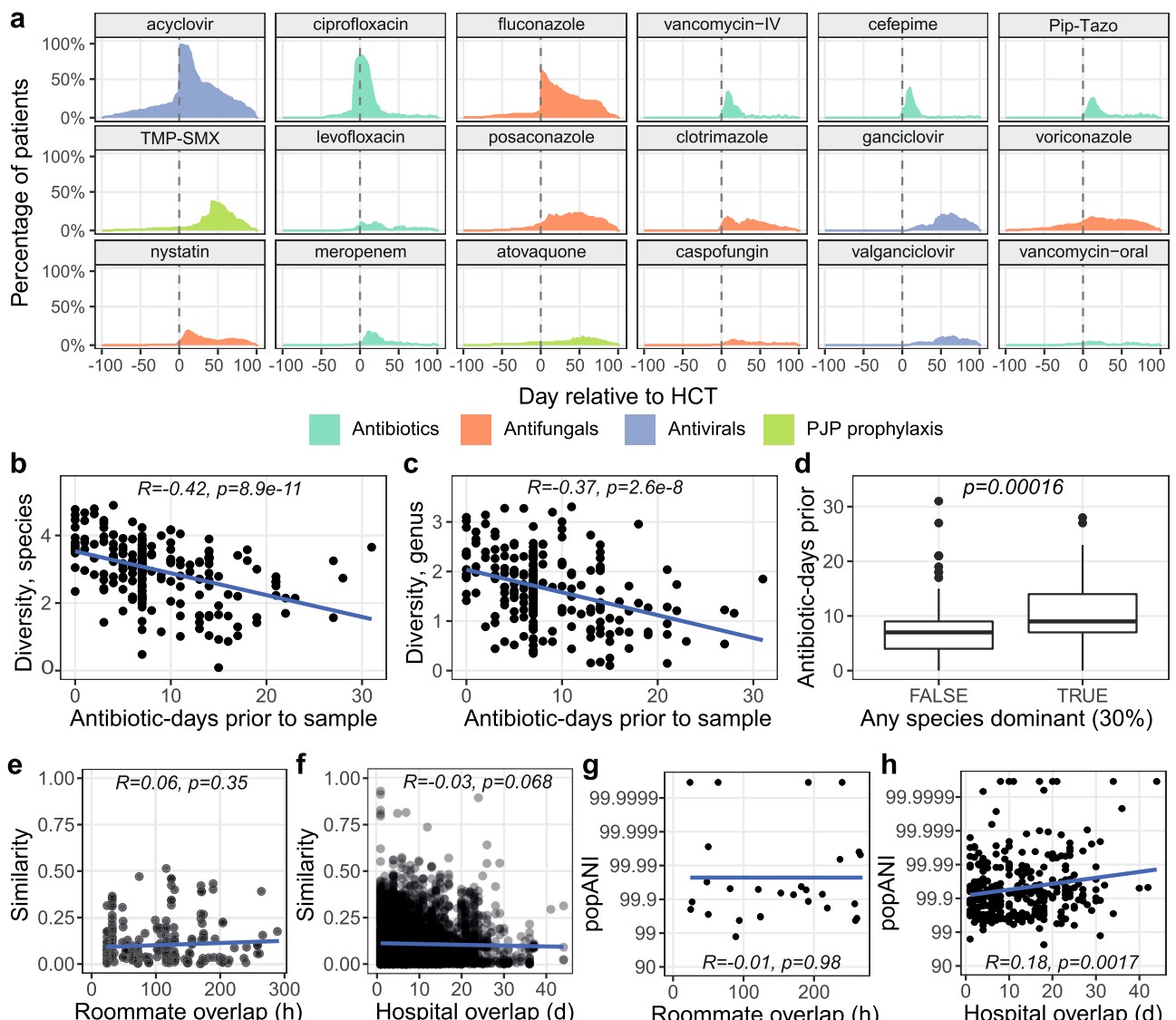

**Fig. 2 The impact of antibiotic prescription and geographic overlap on patient microbiomes. a** Aggregated prescription history of 20 of the most frequently prescribed antibiotic, antifungal and antiviral drugs. Each panel shows the percentage of patients who were prescribed a drug at the given day, relative to the date of HCT. Shannon diversity at the species (**b**) or genus (**c**) level compared to total antibiotic-days in the 7 days prior to sample collection. (**d**) Samples with or without a single species dominant (≥30%), compared with total antibiotic-days in the prior seven days. Boxes extend to the first and third quartile, whiskers extend to the upper and lower value within 1.5*interquartile range (IQR) from the box. Outliers are shown as points. A two-sided Wilcoxon rank-sum test was used to compute the p-value. n = 133 samples from 71 patients in FALSE category; n = 81 samples from 60 patients in TRUE category. Taxonomic similarity at the species level (1−Bray−Curtis dissimilarity) between samples from different patients, evaluated against days of hospital overlap (**e**) or hours of roommate overlap (**f**) prior to the earlier sample. Maximum inStrain popANI achieved by comparing all strains in all samples from two patients, evaluated against hours of roommate (**g**) or days of hospital overlap (**h**) prior to the earlier sample. In all panels, trend lines are calculated as the best-fit linear regression between the X and Y variables. R and p-values are the Pearson correlation coefficient and correlation p-value, respectively. Pip-Tazo Piperacillin-Tazobactam, TMP-SMX trimethoprim-sulfamethoxazole, PJP Pneumocystis jirovecii pneumonia. Source Data for this figure are provided in Supplementary Data 4–9.

counts a SNP only if both samples share no alleles. For example, if A/T alleles were found at frequencies of 90/10% and 10/90% in two samples, a consensus SNP would be called because the consensus base is different. A population SNP would not be called because both samples share an A and T allele.

We mapped sequencing reads from all samples against the collection of 1615 unique MAGs and compared strains in samples from different patients with >50% coverage breadth at a depth of five reads (see methods, Supplementary Data 9). This coverage breadth threshold ensures ANI is calculated across the majority of the strains being compared and is recommended by the authors of inStrain. The maximum log-scaled popANI value was taken as

representative of the maximal strain sharing between all pairs of patients. Length of hospital overlap was significantly associated with having a more similar microbiome strain (linear regression on log-scaled popANI values, p = 0.0017, Fig. 2g). Time of roommate overlap was not significantly associated (Fig. 2h), but the limited number of patients who were roommates in our dataset may preclude us from detecting an association. Given that the taxonomic composition of patient microbiomes does not converge during periods of roommate or hospital overlap (Fig. 2e, f) and that the majority of compared strains are less similar than the 99.999% popANI strain sharing threshold discussed below, this correlation likely does not indicate frequent patient–patient

or hospital–patient transmission. It may represent a subset of similar strains that commonly colonize patient microbiomes upon extended time spent in the hospital. We followed up on the few cases of high-identity strains as evidence for possible transmission events.

**HAI organisms that colonize HCT patient microbiomes are part of known, antibiotic-resistant and globally disseminated clades.** E. coli and E. faecium are common commensal colonizers of human microbiomes[49–51]. These species can also be pathogenic and contribute to inflammation, dysbiosis, and infection in the host[52,53]. The specific strain of these species is key in determining the balance between a healthy and diseased state in the microbiome. We compared patient-derived E. coli and E. faecium MAGs with several reference genomes (Supplementary Table 2, Supplementary Data 10) to identify the closest strains or sequence types.

Escherichia coli. The 95% identity, or species-level, cluster of E. coli MAGs contained 47 genomes from 26 patients (Fig. 3). Within this alignment average nucleotide identity (ANI) based tree, we observed two clades of genomes where MAGs from multiple patients had >99.9% ANI. These clades were investigated further as they may represent common sequence types. The first clade contained 15 MAGs from 7 patients; these MAGs had >99.9% ANI to pathogenic E. coli sequence type (ST) 131 clade C2 genomes, including EC958[54] and JJ1886[55]. ST131 is an extraintestinal, pathogenic, multidrug-resistant E. coli strain which frequently causes urinary tract infections[56]. E. coli ST131 often carries extended-spectrum β-lactamase (ESBL) genes which convey a wide range of antibiotic resistance. This sequence type is believed to colonize the intestinal tract even in healthy individuals without antibiotic exposure[57] and there are reports of this pathogen causing urinary tract infections in multiple individuals within a household[58].

The second clade contained 12 MAGs from 5 patients with >99.8% ANI to the pathogenic ST648 representative IMT16316[59]. ST648 is also an ESBL-producing E. coli strain, but it is not as widespread as ST131. Both STs have been isolated from wastewater[60] and ST648 has been isolated from the gut of humans[61] and other mammals[62]. Our finding that E. coli ST648 is also prevalent in HCT patient microbiomes suggests that it may become a pathogen of interest in this patient population in the future.

To understand the antibiotic resistance capabilities of the E. coli strains colonizing these HCT patients, we searched for β-lactamase genes in the E. coli MAGs (see the "Methods" section). The most commonly detected genes were ampH and ampC, which are part of the core E. coli genome and likely do not contribute to antibiotic resistance[63] (Fig. S2A). CMY-132 was detected exclusively in MAGs in the ST131 clade, and mutations conveying resistance in gyrA were detected in MAGs in multiple clades. CTX-M-type β-lactamases were detected in several samples, but often within the metagenome rather than within the E. coli MAG, indicating they may be on plasmids or mobile genetic elements that did not bin with the rest of the E. coli genomes.

Enterococcus faecium. The species-level cluster of E. faecium MAGs contained 30 genomes from 20 patients (Fig. 4a). All MAGs were ≥99% identical, suggesting a single ST is present in most patients[50]. These genomes matched closest to E. faecium ST117, a well-described vancomycin-resistant strain that frequently causes BSIs[64]. Notably, five other MAGs had ~94% ANI (below the 95% clustering threshold, and therefore not shown in

Fig. 3a) to the ST117 clade and >99% ANI to commensal E. faecium strains, including strains Com15 and Com12[50]. To understand the vancomycin resistance capabilities of these strains, we searched for the seven genes in the vanA operon[65]. Only 2/30 samples had the full operon present within the E. faecium MAG (Fig. S2b). When we looked in the entire metagenome, 22/30 samples had the full operon present, and the vanA genes were usually detected on a contig that was not assigned to any MAG. Van genes are often carried on mobile genetic elements or plasmids in E. faecium, which frequently do not bin well.

Interestingly, no van genes are present in sample 11342_02, but the full operon is present in the E. faecium MAG in sample 11342_03, collected 14 days later from the same patient. In sample 11342_03, the vanA operon appears on a contig 8.7 kb long with 143× coverage, much lower than the ~2000× coverage of the rest of the E. faecium MAG. Mapping reads against this contig revealed scattered coverage in 11342_02, leading us to believe the operon is present, but not at high enough coverage to be assembled in the earlier sample. We attempted to culture vancomycin-resistant Enterococcus (VRE) from these samples (see the "Methods" section), and positively identified E. faecium by MALDI coupled to time-of-flight mass spectrometry in samples 11342_02 and 11342_03. Taken together, these results suggest that there were at least two strains of E. faecium in the gut microbiota of patient 11342. The low relative coverage of the vanA operon indicates that the VRE strain may have been a small fraction of the total E. faecium pop ulation. Patient 11342 was never prescribed vancomycin (Fig. S7a), so the VRE strain may not have had an advantage in this environment.

VanA genes were also not detected in sample 11349_01, where we observed a nearly identical E. faecium strain to patient 11342 after the patients shared a room for 11 days. When we attempted to culture VRE from 11349_01, a bacterium grew poorly on plates containing vancomycin and was identified as Klebsiella pneumoniae. Therefore, we believe the E. faecium strain in the microbiome of this patient was vancomycin sensitive. If transmission from patient 11342 was responsible for colonization of patient 11349, the vancomycin-sensitive strain may have been transmitted.

**Nearly identical strains indicative of putative patient–patient Enterococcus faecium, but not Escherichia coli transmission.** We used the results from the inStrain comparison to search for nearly identical bacterial strains, which may be indicative of transmission between patients. To determine a threshold for putative transmission, we examined comparisons in several "positive control" datasets where we expect to find identical strains, either as the result of persistence or transmission: time-course samples from the same HCT patient, stool samples from mother–infant pairs[18] and samples from fecal microbiota transplantation donors and recipients[23]. We often observed 100% popANI in these "positive control" comparisons, indicating that there were no SNPs that could differentiate the strain populations in the two samples (Figs. S3 and S4). Due to expected noise and errors in sequencing data, we set the lower bound for transmission in our HCT cohort at 99.999% popANI, equivalent to 30 population SNPs in a 3 megabase (Mb) genome. The same threshold was used to identify identical strains by the authors of inStrain[48].

Escherichia coli. While E. coli genomes in samples collected from the same patient over time were always more similar than the putative transmission threshold, in no case did we observe a pair of samples from different patients with ≥99.999% popANI

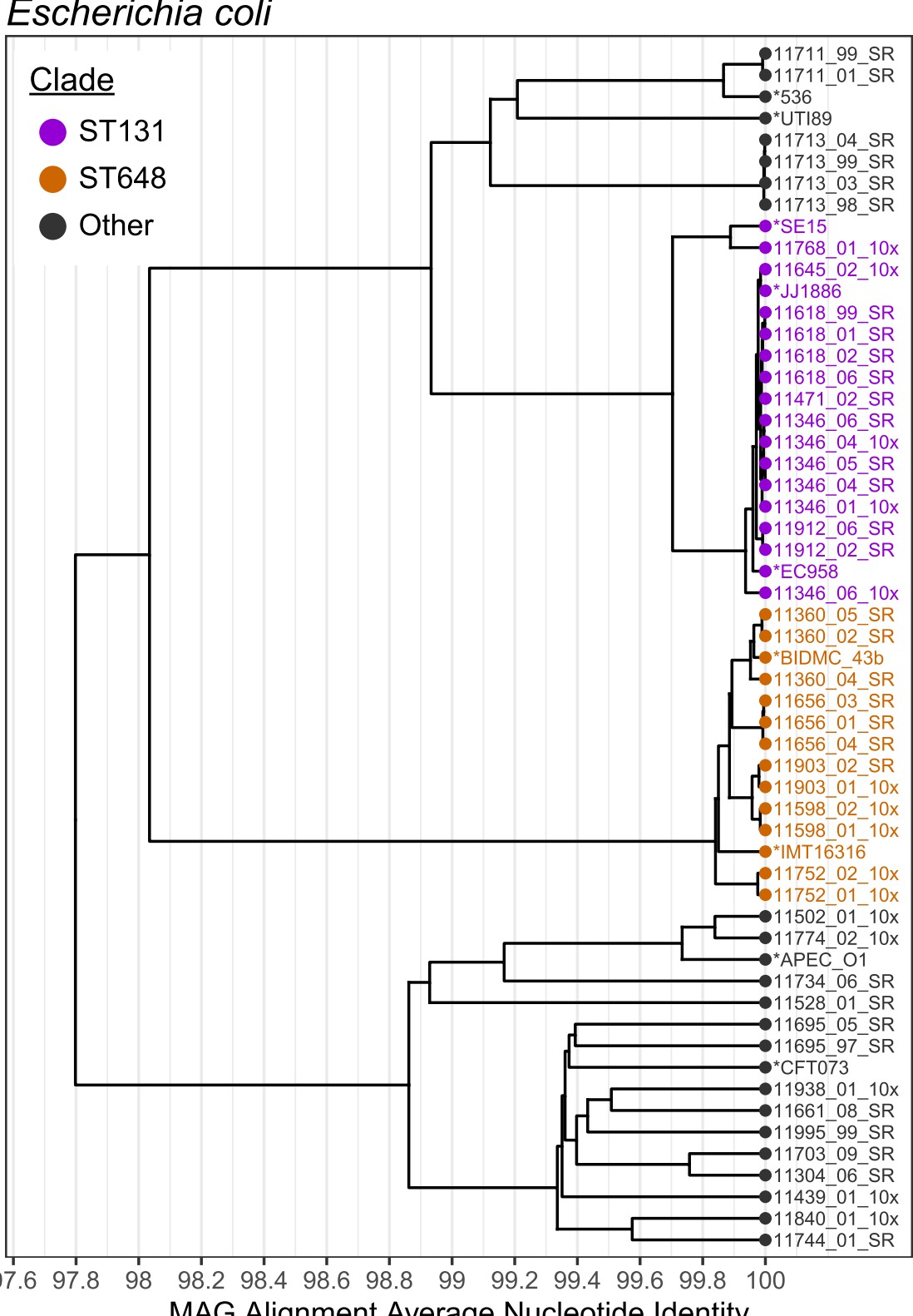

**Fig. 3 Alignment average nucleotide identity (ANI) tree of *Escherichia coli* MAGs.** MAGs identified as *E. coli*, medium quality or above and at least 75% the mean length of the reference genomes are included. Several reference genomes are included and labeled with an asterisk. Clusters at the 99% ANI level corresponding to ST131 (purple) and ST648 (orange) are highlighted. Source Data for this figure are provided in Supplementary Data 10.

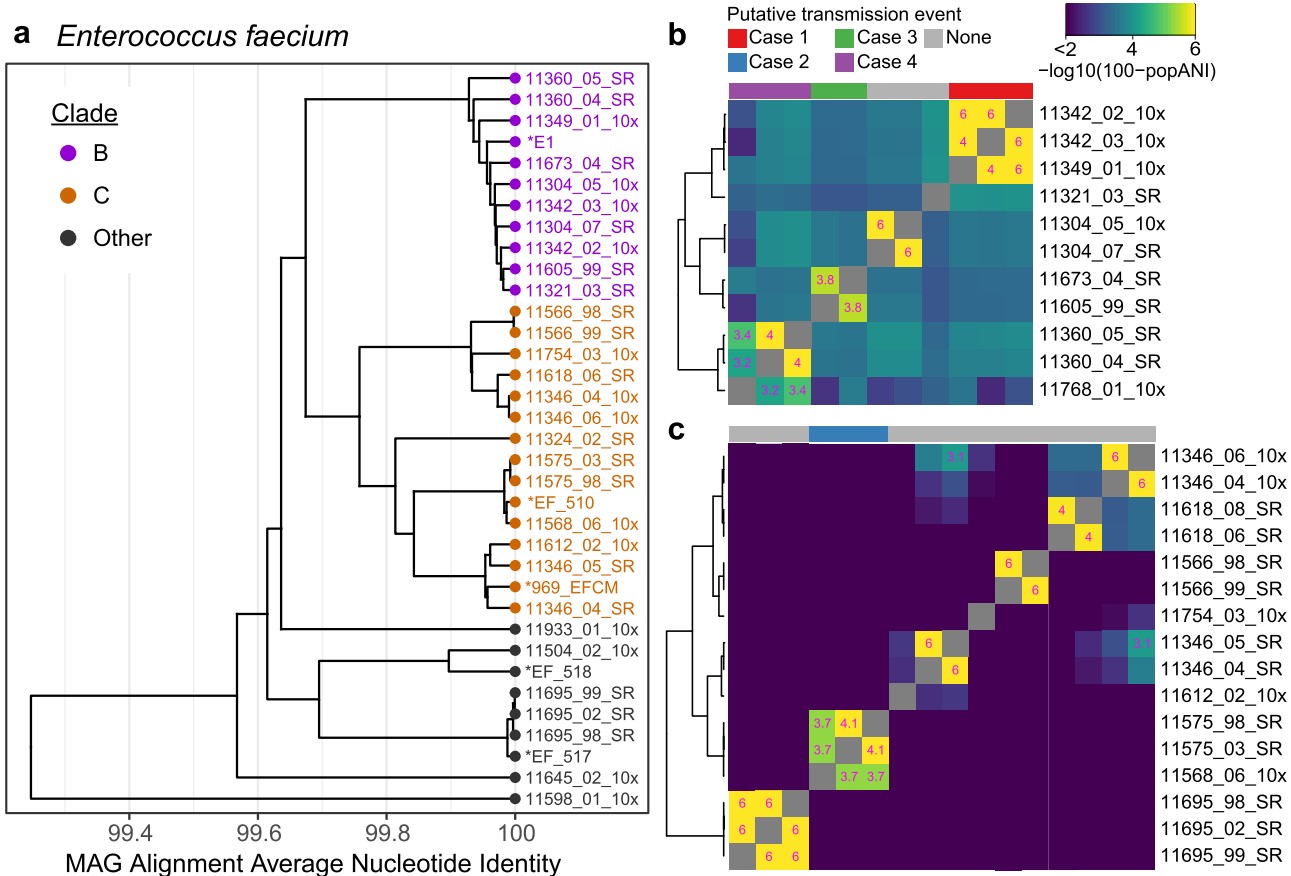

**Fig. 4 *Enterococcus faecium* strains compared between patients. a** Alignment average nucleotide Identity (ANI) based tree of *E. faecium* MAGs. MAGs identified as *E. faecium*, medium quality or above and at least 75% the mean length of the reference genomes are included. Several reference genomes are included and labeled with an asterisk. Two clades containing samples from multiple patients are highlighted for further comparison. Alignment values used to construct this tree can be found in Supplementary Data 10. **b, c** Heatmaps showing pairwise popANI values calculated with inStrain for clades B and C, respectively. Color scale ranges from 99.99–100% popANI and is in log space to highlight the samples with high popANI. Cells in the heatmap above the transmission threshold of 99.999% popANI are labeled. Four groups containing samples from multiple patients with popANI values above the transmission threshold are highlighted on the top of the heatmaps. Source Data for this figure are provided in Supplementary Data 9 and 10.

(Supplementary Data 9). This result suggests that all *E. coli* strains observed are patient-specific and argues that there are not common strains circulating in the hospital environment or passing between patients. Alternatively, patient–patient transmission or acquisition of common environmental strains is either notably rare or rapid evolution after a patient acquires a new strain is reducing popANI levels below the threshold. Deeper metagenomic sequencing or isolation and sequencing of *E. coli* strains may allow us to detect transmission in previously missed cases.

Enterococcus faecium. We performed the same analysis in *E. faecium* and observed four examples where two patients shared a strain with ≥99.999% popANI (Fig. 4b, c). In one case, the two patients were roommates and direct transmission appears to be the most likely route. In the other three cases, epidemiological links were less clear, suggesting the patients may have acquired a similar strain from the hospital environment or through unsampled intermediates. In the following descriptions, samples are referred to by the day of collection, relative to the first sample from patients in the comparison.

### Case 1

Patients 11342 and 11349 overlapped in the hospital for 21 days and were roommates for 11 days (Fig. 5a). Patient 11342 had a gut microbiome that was dominated by *E. faecium*; the two samples from this patient have 60% and 87% *E. faecium* relative abundance. A

single sample from patient 11349 was obtained 14 days after starting to share a room with patient 11342. This sample is dominated by *Klebsiella pneumoniae*, and *E. faecium* is at 0.4% relative abundance. InStrain comparisons between the *E. faecium* strains in 11342 (the presumed "donor") and 11349 (the presumed "recipient") of the strain revealed 0-2 population SNPs (popANI 100–99.9999%) with 87% of the reference MAG (2.24 Mb) covered ≥5× in both samples. MAGs from each patient were also structurally concordant (representative dotplots in Fig. S6a). These genomes were the most similar out of all *E. faecium* genomes compared from different patients. Samples from these patients were extracted in different batches and sequenced on different lanes, minimizing the chance that sample contamination or "barcode swapping"[66] (see Supplementary Methods) could be responsible for this result. No other strains were shared between these two patients.

### Case 2

Patients 11575 and 11568 overlapped in the hospital for 36 days but were never roommates (Fig. 5b). Samples from patient 11575 span 97 days, during which this patient experienced a BSI with *Klebsiella pneumoniae* and treatment with intravenous (IV) vancomycin, ciprofloxacin, meropenem (Fig. S7c). Antibiotic treatment likely resulted in a reduction in microbiome diversity and domination by *E. faecium* in samples collected on days 16 and 28. Two samples were collected from patient 11568 on days 28 and 119. The first sample from 11568 was also dominated by *E. faecium*, but strains from the two patients were distinct (99.95% popANI). 91 days later, the second sample from 11568 has a lower relative abundance of *E. faecium* but a nearly identical strain to patient 11575. Five population SNPs (99.9997% popANI) were detected with 88% of the reference MAG covered ≥5× in both samples (representative dotplot in Fig. S6b). This suggests that the *E. faecium* strain in 11568 was replaced by a different strain with high identity to the strain in 11575. Patient 11568 was discharged from the HCT ward during the period between the two samples. The shared strain may represent an acquisition from a common environmental source or transmission from unobserved patients, rather than a direct transmission event between these two patients. While we observed different *E. faecium*

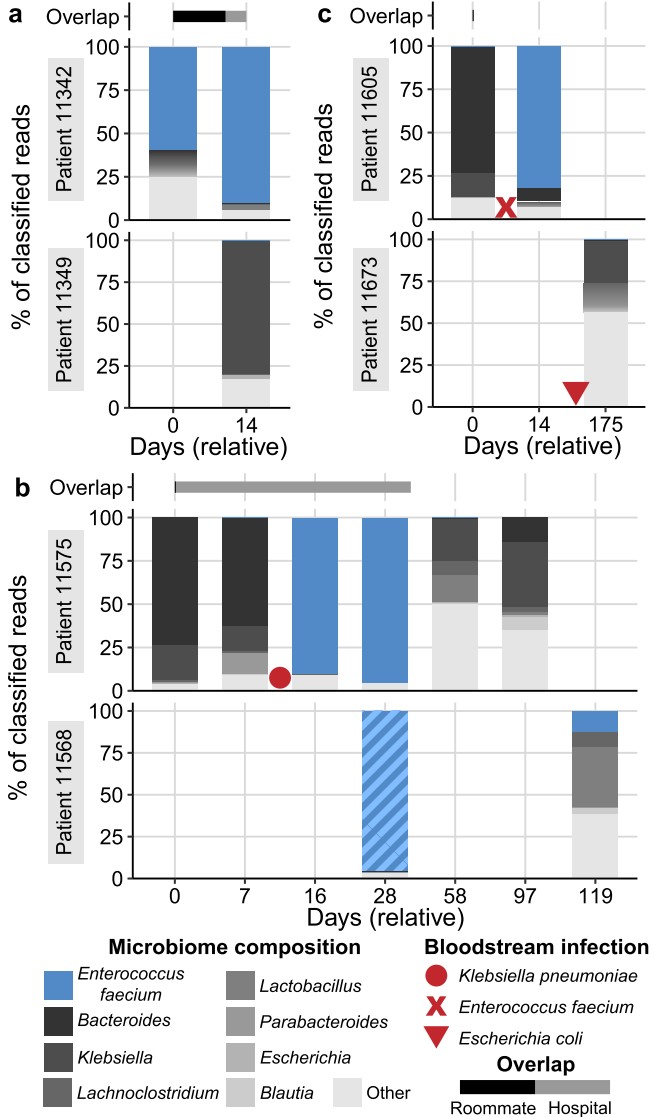

**Fig. 5 Microbiome composition of patients with putative *Enterococcus faecium* transmission events.** Each panel shows the composition of two patients over time. The height of each bar represents the proportion of classified sequence data assigned to each taxon. Samples are labeled relative to the date of the first sample in each set. Bars above each plot represent the approximate time patients spent in the same room (black bars) or in the hospital (gray bars). Red symbols indicate approximate dates of bloodstream infection with the specified organism. Postulated direction of transmission progresses from the top to the bottom patient. Fractions of the bar with >99.999% popANI strains in each panel are indicated with solid colors, and different strains are indicated with hashed colors. All taxa except *E. faecium* are shown at the genus level for clarity. **a** Case 1: Putative transmission from patient 11342 to 11349. **b** Case 2: Putative transmission from patient 11575 to 11568. **c** Case 3: Putative transmission from patient 11605 to 11673. Source Data for this figure are provided in Supplementary Data 7–9.

strains in the two samples collected from patient 11568, an *E. faecalis* strain remained identical between the two time points (99.9997% popANI).

## Case 3

Patients 11605 and 11673 did not overlap in the hospital (Fig. 5c). Two samples were collected from 11605 on days 0 and 14. This patient experienced a BSI with *E. faecium* and treatment with meropenem (Fig. S7e) prior to a sample dominated by the same species on day 14. Patient 11673 experienced a BSI with *E. coli* and treatment with

vancomycin, meropenem, and cefepime (Fig. S7f) prior to the single sample we collected from this patient. Comparing *E. faecium* strains between the two patients revealed 2 population SNPs (99.9998% popANI) with 48% of the reference MAG covered ≥5× in both samples (representative dotplot in Fig. S6c). Although slightly below the 50% coverage threshold, the high degree of similarity caused us to consider this result. While *E. faecium* strains in the two patients were nearly identical, the samples were collected 161 days apart and the patients had no overlap in the hospital. This suggests both patients may have acquired the strain from the hospital environment, through transmission from unsampled patients, or another source such as healthcare workers.

## Case 4

Patients 11360 and 11789 did not overlap in the hospital. *E. faecium* remained at relatively low abundance in all samples. Comparing *E. faecium* strains between patients revealed 5–10 population SNPs (99.9993–99.9996% popANI) with 50–57% genome coverage. Neither patient had a BSI during the sampling period. As these samples were collected at least 428 days apart, a shared source again may be the most likely explanation.

*Comparisons with* E. faecium *and* E. coli *in published data.* The *E. faecium* and *E. coli* strains we observe in our patients may be unique to this patient population and hospital environment. Alternatively, they may be hospital acquired strains that are present in other settings around the globe. We searched through several published datasets to differentiate between these possibilities. Our comparison dataset included metagenomic shotgun sequence data from 189 stool samples from adult HCT patients[67], 113 stool samples from pediatric HCT patients[3,68,69], 732 stool samples from hospitalized infants[70] and 58 vancomycin-resistant *E. faecium* isolates[71]. Sequence data were downloaded from SRA and processed in the same manner as other short-read data. Each sample was aligned against the *E. faecium* and *E. coli* MAGs used in the inStrain analysis above, profiled for SNPs, and compared against samples collected from our HCT patients. Comparisons within our data and comparisons within individual external datasets frequently achieved popANI values of ≥99.999%, typically from comparisons of samples from the same patient over time. Meanwhile, comparisons between our samples and external samples had lower popANI values (Fig. S5).

Comparisons of *E. faecium* strains in samples from patient 11346 in our dataset and patient 688 in the HCT microbiome dataset collected at Memorial Sloan Kettering Cancer Center[67] demonstrated a maximum of 99.9993% popANI (16 population SNPs detected in 2.3 Mb compared). While direct transmission is likely not involved here, this observation does align with the nearly identical *E. faecium* strains we observed in patients with no hospital or roommate overlap (cases 3 and 4) and speaks to the global dissemination of vancomycin-resistant *E. faecium* ST 117. Comparing *E. coli* to external datasets revealed a maximum similarity of 99.996% popANI (200 population SNPs detected in 5.0 Mb compared).

**Putative transmission of commensal bacteria.** Next, we extended the inStrain analysis to compare all species that were present in multiple patients. We found nearly identical genomes of commensal organisms that may be the result of transmission between patients, as well as several species shared between patients without clear explanations.

*Hungatella hathewayi.* Patients 11639 and 11662 overlapped in the hospital for 34 days and were roommates for a single day, after which 11639 was discharged (Fig. 6a). *Hungatella hathewayi* was at 5–10% relative abundance in the two samples from 11639. Patient 11662 developed *Streptococcus mitis* BSI on day 10 and was treated with IV vancomycin and cefepime (Fig. S8b). The microbiome of this patient recovered with markedly different composition, including an abundant *H. hathewayi* strain reaching 54% and 17% relative abundance on days 58 and 100, respectively. Comparing *H. hathewayi* genomes between these two

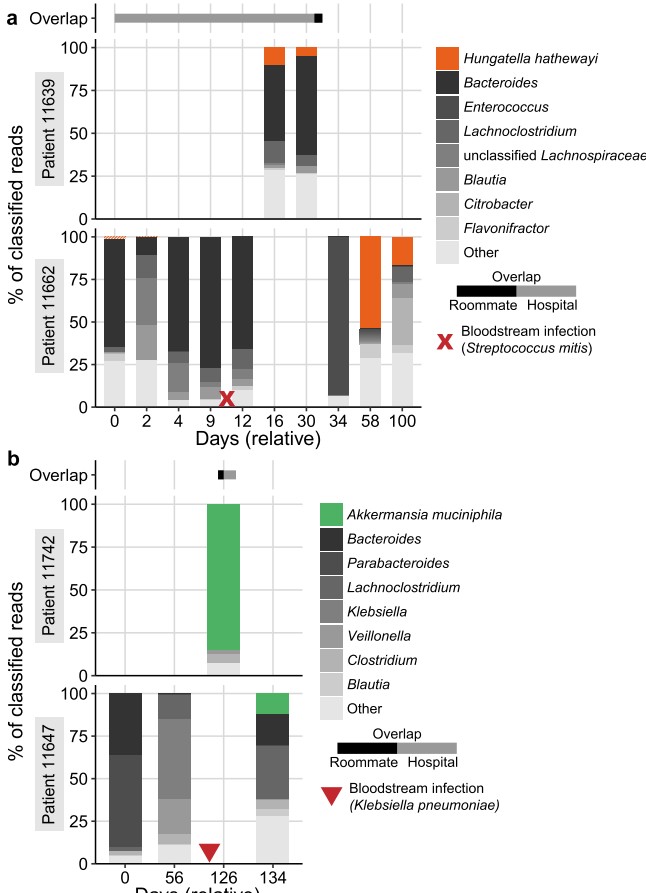

**Fig. 6 Microbiome composition of patients with putative *Hungatella hathewayi* or *Akkermansia muciniphila* transmission events.** Each panel shows the composition of two patients over time. The height of each bar represents the proportion of classified sequence data assigned to each taxon. Samples are labeled relative to the date of the first sample in each set. Bars above each plot represent the approximate time patients spent in the same room (black bars) or in the hospital (gray bars). Red symbols indicate approximate dates of bloodstream infection with the specified organism. Hypothesized direction of transmission progresses from the top to the bottom patient. Fractions of the bar with >99.999% popANI strains in each panel are indicated with solid colors, and different strains are indicated with hashed colors. All taxa except *H. hathewayi* or *A. muciniphila* are shown at the genus level for clarity. **a** Putative case of *H. hathewayi* transmission from 11639 to 11662. **b** Putative case of *A. muciniphila* transmission from 11742 and 11647. Source Data for this figure are provided in Supplementary Data 7–9.

patients revealed 0–1 population SNPs (100–99.99998% popANI) with 94–98% coverage ≥5× (6.9–7.1 Mb sequence covered in both samples). This was the single highest ANI comparison among all strains shared between patients. *H. hathewayi* MAGs from these patients were also structurally concordant and had few structural variations (Fig. S6d). No other strains were shared between these patients.

Patient 11662 had *H. hathewayi* in the first two samples at 1.2% and 0.3% relative abundance, respectively. Although we were limited by coverage, comparing early to late samples with inStrain revealed 472 population SNPs in 3% of the genome that was covered at least 5×, implying 11662 was initially colonized by a different *H. hathewayi* strain, which was eliminated and subsequently replaced by the strain present in 11639. Given that samples were collected weekly, determining the direction of

transmission is challenging. However, 11639 to 11662 appears to be the most likely direction, given the sampling times and perturbation 11662 experienced. However, it is possible that transmission occurred in the opposite direction or from a common source. Interestingly, 11662 was also re-colonized with *Flavonifractor plautii* in later samples. This strain was different from the strain in earlier samples from this patient, as well as all other *Flavonifractor* strains in our sample collection.

*H. hathewayi* is known to form spores and is able to persist outside a host for days[72]. Although these patients were only roommates for a single day, 11662 remained in the same room for 4 days after 11639 was discharged, increasing the chance that a *H. hathewayi* spore could be transmitted from a surface in the shared room or bathroom. The question remains as to why transmission of *H. hathewayi* is not more common, given it is found at ≥1% relative abundance in 31 patients. Perhaps the earlier colonization of the microbiome of 11662 with a different *H. hathewayi* strain was key—the microbiome in this patient was "primed" to receive a new strain of the same species, despite the significant perturbation this patient experienced.

Notably, *H. hathewayi* was recently reclassified from *Clostridium hathewayi*[73], and was previously shown to induce regulatory T-cells and suppress inflammation[74]. Although the interaction of this microbe with the HCT process is not known, it may be interesting to investigate further given that the microbe may be transmitted between individuals and may contribute to inflammation suppression that may be relevant in diseases such as graft-vs.-host disease. However, *H. hathewayi* may not be entirely beneficial or harmless and has been reported to cause BSI and sepsis in rare cases[75,76].

*Akkermansia muciniphila*. Patients 11742 and 11647 overlapped in the hospital for 11 days and were roommates for nine days (Fig. 6b). Patient 11647 experienced a BSI with *Klebsiella pneumoniae* (perhaps related to previous *K. pneumoniae* domination of the microbiome) and was treated with piperacillin-tazobactam and cefepime (Fig. S8d). The final sample from 11647 has *Akkermansia muciniphila* at 9.4% relative abundance, while the single sample from 11742 was dominated by *A. muciniphila* (85% relative abundance). Comparing these genomes revealed 0 population SNPs and 7 consensus SNPs with 90% coverage, as well as concordant MAGs from each sample (Fig. S6e). No other strains were shared between these two patients.

In contrast to *H. hathewayi*, *A. muciniphila* is not known to form spores, which may reduce the chance of this microbe being transmitted. However, it is an aerotolerant anaerobe that may survive in oxygen for short periods of time[77]. The microbiome domination of 11742 with *A. muciniphila* and the relatively long overlap period of 9 days in the same room may provide a greater "infectious dose" (abundance*exposure time) to the recipient patient.

**Widespread strain sharing of commercially available probiotic organisms.** Several organisms were found with identical or nearly identical genomes across multiple patient microbiomes without clear epidemiological links. The largest clade was found for *Lactobacillus rhamnosus*, which included 11 samples collected from eight patients over a span of 2.5 years (Fig. 7a, d). All 11 samples in this clade had pairwise popANI of ≥99.999%, and in a subset of eight samples from seven patients, all pairs were identical from a popANI perspective (100%). Of the eight patients in this clade, only two pairs were roommates or overlapped in the hospital (patients 11537/11547 and 11647/11662, roommates for three days and one day, hospital overlap for 20 and 44 days, respectively). All 11 samples in this clade were collected after

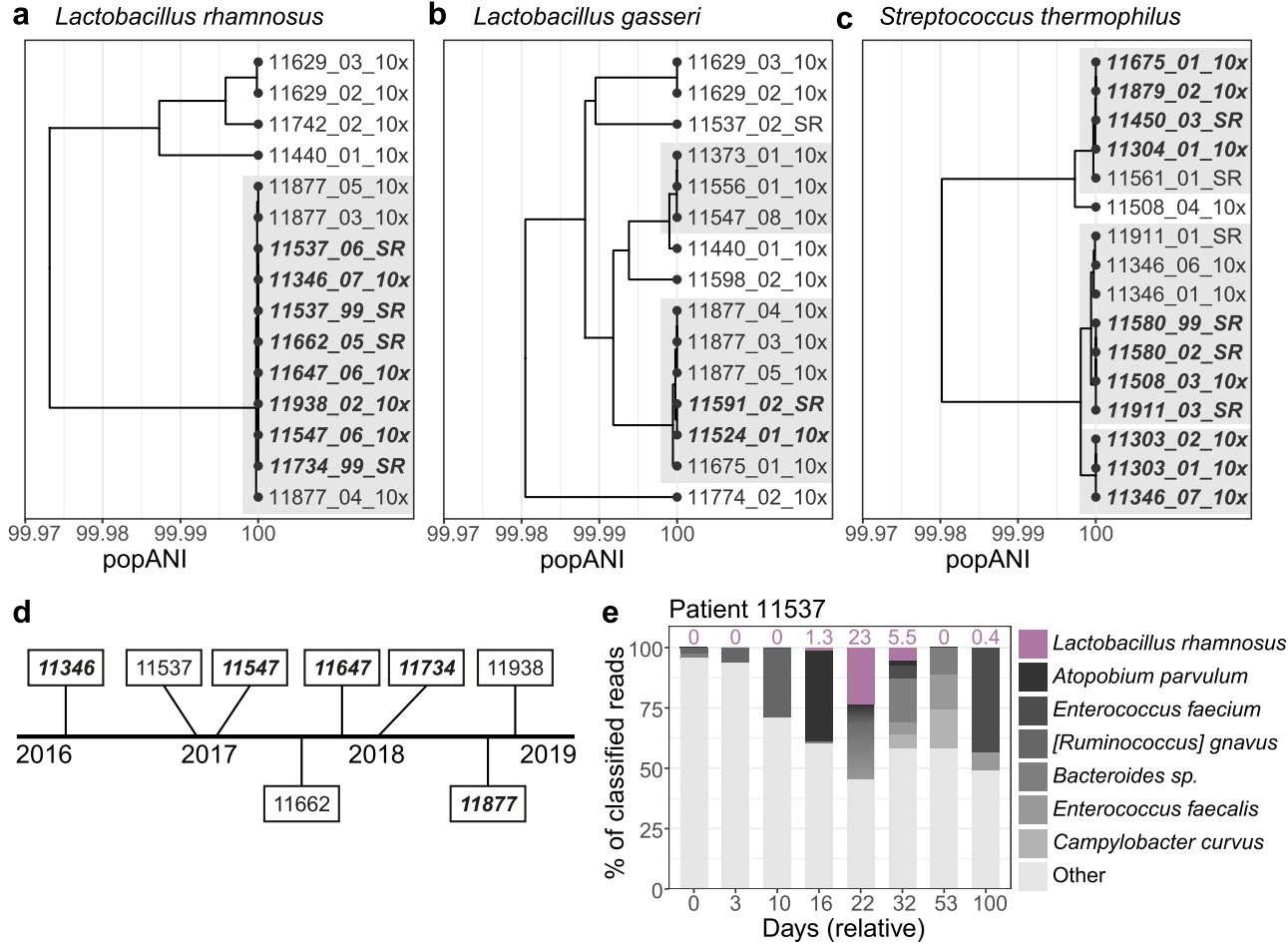

**Fig. 7 _Lactobacillus_ and _Streptococcus_ strains are acquired after HCT and identical between many patients.** Population ANI-based tree of (**a**) _Lactobacillus rhamnosus_, (**b**) _Lactobacillus gasseri_, (**c**) _Streptococcus thermophilus_ strains present in patient samples. Clades containing samples from different patients with ≥99.999% popANI are highlighted with a gray background. Clades with 100% popANI between all pairs are additionally bolded and italicized. **d** Timeline of approximate date of samples containing a _L. rhamnosus_ strain in the transmission cluster in (**a**). Patients who were discharged from the hospital after HCT and prior to acquiring _L. rhamnosus_ are bolded and italicized. **e** Microbiome composition of patient 11537. _L. rhamnosus_ abundance at each time point is indicated above the bar. This patient received HCT on relative day 3. Source Data for this figure are provided in Supplementary Data 7–10.

HCT (median time from HCT to first sample 43 days, range 12–93 days), and _L. rhamnosus_ always had <0.1% relative abundance in pre-HCT samples, when samples were available. Five of eight patients were discharged from the hospital after HCT and prior to acquiring _L. rhamnosus_, which we observed in a sample collected during a subsequent admission. We also observe _L. rhamnosus_ falling below 0.1% relative abundance in a subsequent sample in five patients, suggesting that this strain may be a transient colonizer of the microbiome (Fig. 7e). We evaluated antibiotic prescriptions in these patients and found that acquisition and loss of _L. rhamnosus_ typically occur independently of antibiotic use. For example, in patient 11537, _L. rhamnosus_ is first detected and expands to 23.4% relative abundance while the patient is prescribed ciprofloxacin and cefepime. _L. rhamnosus_ declines in relative abundance after the antibiotic prescription ends (Fig. S9). Similar clades of high-identity genomes from different patients were found for _Lactobacillus gasseri_ (Fig. 7b) and _Streptococcus thermophilus_ (Fig. 7c).

Given that we did not observe hospital or roommate overlap between most patients in the _L. rhamnosus_ cluster, the most likely explanation is that patients acquired this strain from a common source. _L. rhamnosus_ is a component of several commercially available probiotic supplements, is present in certain live active-culture foods such as yogurt, and is among the most commonly

prescribed probiotic species in US hospitals[78]. However, HCT recipients were not allowed to take probiotics or consume high-bacteria dairy products, such as probiotic yogurt or soft cheese, while inpatients on the HCT ward. We also verified that no prescriptions were written for probiotics by examining electronic health records. A majority of patients were discharged from the hospital between HCT and acquiring the _L. rhamnosus_ strain, which may have provided them with the opportunity to consume a probiotic supplement or dairy product. Contact with a family member or other individual who had the strain in their microbiome could also be responsible for colonization of the HCT patient.

If this _L. rhamnosus_ strain is a commonly used probiotic supplement or is found in commonly consumed dairy products, it may be found in other gut microbiome sequencing datasets. Comparing MAGs from this cluster against all Genbank genomes revealed a maximum alignment-based ANI of 99.95% to _L. rhamnosus_ ATCC 8530[79]. Instrain-based comparisons against this reference had a maximum popANI of 99.98%, below the putative transmission threshold. We then searched against all genomes in the Unified Human Gastrointestinal Genome collection[41] and identified two genomes that were nearly identical to the strain found in HCT patients. These genomes were originally from the Human Gastrointestinal Bacteria Culture

Collection[80] (accessions ERR2221226 and ERR1203919, belonging to the same isolate per a personal communication with the authors). Assembled isolate and patient-derived genomes had ≥99.99% ANI; inStrain-based SNP comparisons had ≥99.999% popANI. This suggests that a *L. rhamnosus* strain that is nearly identical to the genomes in our HCT patients has been isolated from human stool in the past.

## Discussion

Our investigation using high-resolution metagenomic sequencing attempts to quantify if and when patient-patient microbiome transmission is involved in the spread of pathogenic organisms. We first found that hospitalized HCT patients frequently harbor HAI organisms in their gut microbiome, validating previous studies which used culture-based approaches or 16S rRNA sequencing[6,26,27]. MAGs created from patient samples had a high identity to several globally disseminated and antibiotic-resistant sequence types, including *Escherichia coli* ST131 and ST648. Interestingly, whereas ST131 is a well-recognized multi-drug resistant pathogen, ST648 is nearly as prevalent as ST131 in our sample collection, and may thus be an emerging pathogen in this patient population.

Despite the high degree of hospital and roommate overlap between patients in our study, we found no association between these factors and the taxonomic similarity of patient microbiomes, and only a weak association between hospital overlap and maximum strain identity. These findings suggest that patient-patient transmission is not driving microbiome composition, but individual strains may still be shared between patients. We did not identify any pairs of patients harboring *E. coli* strains with popANI values above the 99.999% popANI transmission threshold. Taken together with the observation that *E. coli* is commonly detected in the patient's microbiome upon admission, this finding argues that patients usually enter the hospital with an "individual-specific" *E. coli* strain and do not frequently transmit it to others. An exclusion principle may be at play, where an *E. coli* niche can only be filled by a single strain or a small number of strains, and new strains are unlikely to engraft when the niche is already occupied. In contrast to *E. coli*, we observed four pairs of patients with *E. faecium* strains that were more similar than 99.999% popANI. In one case, the two patients spent 11 days sharing a room and bathroom prior to observing the shared strain. Direct links between patients were less clear or nonexistent in the other three cases, and transmission through unsampled intermediates or acquisition from an environmental source may have been responsible. We also found evidence for an *E. faecium* strain in a published dataset from HCT patient microbiomes[67] that had above 99.999% popANI to a strain in our sample collection. While this finding is likely not the result of patient-patient transmission, it does indicate that very similar strains may exist within patients in different geographic locations.

We then expanded the transmission analysis to examine all species that were present in the microbiome of multiple patients. Identical *Hungatella hathewayi* strains were found in two patients who were in the hospital together for 34 days and roommates for a single day. Earlier samples from patient 11662 had a significantly different *H. hathewayi* strain than the strain present in later samples. It is possible that the earlier colonization with the same species exhibited a priority effect[81] and primed this individual to be re-colonized. In another set of patients who overlapped in the hospital for 11 days and were roommates for nine days, we identified identical *Akkermansia muciniphila* strains. In both examples, the likely "recipient" patient experienced BSI prior to the putative transmission event. The subsequent antibiotic treatment initiated for the treatment of BSI resulted in vast microbiome modification and simplification, which may have opened a niche for the new organism to engraft into. Both *H. hathewayi* and *A. muciniphila* can survive outside the host for periods of time, but *H. hathewayi* can form spores that enable it to live in aerobic conditions for days[72]. In the case of *H. hathewayi* transmission, patient 11662 remained in the same room with a shared bathroom after the single day of overlap with patient 11639. Spores surviving on surfaces may be responsible for transmission, given the relatively short period of overlap. In these cases, patient-patient transmission may help in the recovery of microbiome diversity following BSI and may play a role in ameliorating post-HCT inflammatory processes, such as acute graft-vs-host disease.

Finally, we observed identical probiotic species in multiple patients without clear geographic or temporal links, including *Lactobacillus rhamnosus*, *Lactobacillus gasseri*, and *Streptococcus thermophilus*. Acquisition from a commercially available probiotic or live-active culture food appears to be the most likely explanation. While patients hospitalized for HCT were not allowed to consume probiotics or high-bacterial dairy foods, a majority of patients were discharged after HCT and prior to a subsequent admission, upon which *L. rhamnosus* was detected. Many patients lost the strain in later samples, independent of antibiotic prescription, suggesting that *L. rhamnosus* was a transient colonizer. This matches the observation that abundance of probiotic species in the gut often declines after supplementation ends[20].

The healthy adult gut microbiome is relatively resistant to perturbation and colonization with new strains or species[19]. By contrast, mother-to-infant transmission of bacteria and phages is common and well-described[16–18]. Patients in our study often shared spaces, were exposed to dramatic "niche clearing" therapies and were often immunosuppressed. We frequently observed patients acquiring new organisms into their gut microbiome during their hospital stay, especially following BSI. Still, we found that patient-patient transmission of gut microbes is relatively rare. This suggests that other factors such as age, rather than perturbation or microbial exposure, may play the largest role in microbiome transmission. There are also several alternative explanations for the relative lack of transmission between patients. First, the adult gut microbiome may remain densely colonized even when dramatically perturbed by antibiotics and chemotherapy, and thus may be resistant to invasion with new strains. Strains we observed in later samples may have existed at low very levels in earlier samples from the individual, therefore evading detection. Second, it is possible that the microbiome of healthcare workers, hospital visitors, or other staff serves as the source of newly colonizing strains. Third, it is possible that the built environment, equipment used in the care of these individuals, and other environmental sources such as food and personal items harbored the microbes that were later transmitted. As we did not sample these other potential sources, it is difficult to know the extent to which they contributed to the collective reservoir of potentially transmitted organisms. Fourth, adaptive evolution may rapidly change the genomes of newly acquired microbes, as has been described for *E. faecium* in the human gut[82] and *E. coli* in the mouse gut[83–85]. Rapid evolution would move the genomes of transmitted microbes below the popANI transmission threshold. Finally, transmitted organisms may be killed by antibiotics before they can establish a community within the host and thus be detected with metagenomic sequencing.

Our findings have important implications for hospital management and infection prevention. 55/149 patients (37%) in our study experienced BSIs, which is comparable to the rate of BSI in other transplant centers[86]. Our findings suggest that microbiome transmission does not play a large role in spreading infections

among HCT patients, and that established contact precautions and procedures for patient isolation were working as intended. Recently, the HCT ward at our hospital moved to a new location with exclusively single rooms, which may further reduce the opportunity for transmission.

Our analysis of transmission of microbes between HCT patients does have several limitations. We analyzed hundreds of samples collected over many years, but most sampling was done on a weekly basis. We did not explicitly collect samples on the day of admission or discharge. Our sample collection also ignores previous hospital stays, either in a different ward in our hospital, or other hospitals entirely, that may be responsible for the acquisition of HAI organisms. Our study was focused on collecting fewer samples from a larger number of individuals, potentially limiting our ability analyze microbiome changes over time. By contrast, the second largest study using shotgun metagenomics to study the microbiome of HCT patients had much more dense sampling, analyzing an average of 8 samples from each of the 49 patients in the study[67]. We also did not perform any sampling of the hospital environment, healthcare workers or visitors, which would allow us to track transmission patterns in detail and more conclusively state where newly acquired microbes originated[87]. Our work is also entirely based on metagenomic sequencing data, which has its own challenges and sources of bias, including "barcode swapping"[66], which could contribute to false positive transmission findings. To address this, we measured the impact of barcode swapping in linked-read data, and eliminated linked-read and short-read comparisons where a finding of identical strains could be the result of barcode swapping (see Supplementary Methods). Additionally, metagenomic sequencing may fail to detect lowly-abundant colonizers, especially when samples are contaminated with host DNA[88], which is often the case in stool samples from HCT patients. These challenges undoubtedly affect our sensitivity and specificity in measuring acquisition and transmission of both pathogenic and commensal microbes. While our comparison methods were sensitive to strain populations in the gut microbiome, we did not attempt to phase strain haplotypes. Haplotype phasing with long-read sequencing technology like Nanopore or PacBio[89–91] could help determine whether sets of SNPs occurred in the same or different strains.

Our study leaves several questions unanswered that we hope future work on microbiome transmission will attempt to answer. First, our findings need to be validated in an external cohort in a different hospital. Collecting stool samples during an infection outbreak may lead to more transmission events being identified and may implicate microbiome transmission in perpetuating the outbreak. Similar experiments in a pediatric patient population may reveal more gut-to-gut transmission, as young children have microbiomes that are still developing and more susceptible to colonization with new species. Our work did not investigate any possible sources of microbial transmission other than the gut microbiota of HCT patients. A more granular study where samples are collected from the hospital environment, as well as hand swabs and stool samples from healthcare workers, visitors, and family members, is clearly indicated by these early results. As more transmission events are observed with high-resolution genomic methods, we will start to uncover the general principles governing community assembly in the human microbiome. These new insights may help prevent infections and other illnesses in this vulnerable patient population in the future.

## Methods

**Cohort selection**. HCT patients were recruited at the Stanford Hospital Blood and Marrow Transplant Unit under an IRB protocol approved by The Stanford University Research Compliance Office (Protocol #8903; Principal Investigator: Dr. David Miklos, co-Investigator: Drs. Ami Bhatt and Tessa Andermann). Informed

consent was obtained from all individuals whose samples were collected. Consent included explicit agreement to long-term storage of samples for future research studies and use of health information including but not limited to medication data as well as transplant outcomes. Stool samples were placed at 4 °C immediately upon collection and processed for storage at the same or following day. Stool samples were aliquoted into 2-mL cryovial tubes and homogenized by brief vortexing. The aliquots were stored at −80 °C until extraction.

We identified all samples that had been sequenced previously by our group. Samples were selected for linked-read sequencing to augment this collection. We examined the network of patient roommate overlaps to find cases where we were likely to uncover transmission events, if they were happening. These included patient pairs from whom we ideally had samples before and after the roommate overlap period. 96 samples that provided the best coverage of roommate overlaps were selected for linked-read sequencing.

The following clinical data were extracted from the electronic health record: demographic information, underlying disease, type of transplantation (allogeneic vs. autologous), date and type of bloodstream infection, medication prescriptions, time of admission and discharge and location of patients (rooms) over time. Hospital-wide BSI data were obtained from an electronic report generated by the clinical microbiology laboratory. Medication prescription data was filtered by the following criteria:

1. Only entries for antibiotics, antifungals, antivirals, antibacterials, and *Pneumocystis jirovecii* pneumonia prophylaxis were retained.
2. Medications with a missing start or end date were excluded.
3. Medications with a frequency of "PRN" (*pro re nata*, or *as needed*) or a prescription status of "Canceled" were excluded.
4. Medications with a difference between start and end date of less than one day were excluded.
5. Medications prescribed to be taken by eyes, ears, topical application, or "swish and spit" were excluded.
6. Medication prescriptions occurring outside the window of HCT date ± 100 days were excluded for the aggregated analysis.

We do acknowledge the challenge of working with electronic health record data, and recognize that there is a disconnect between medications prescribed and medications consumed by a patient.

**DNA Extraction, library preparation and sequencing**. DNA was extracted from stool samples using a mechanical bead-beating approach with the Mini-Beadbeater-16 (BioSpec Products) and 1-mm diameter zirconia/silica beads (BioSpec Products) followed by the QIAamp Fast DNA Stool Mini Kit (Qiagen) according to manufacturer's instructions. Bead-beating consisted of 7 rounds of alternating 30 s bead-beating bursts followed by 30 s of cooling on ice. For samples subjected to linked-read sequencing, DNA fragments less than ~2 kb were eliminated with a SPRI bead purification approach[92] using a custom buffer with minor modifications: beads were added at 0.9×, and eluted DNA was resuspended in 50 μl of water. DNA concentration was quantified using a Qubit fluorometer (Thermo Fisher Scientific). DNA fragment length distributions were quantified using a TapeStation 4200 (Agilent Technologies).

Short-read sequencing libraries were prepared with either the Nextera Flex or Nextera XT kit (Illumina) according to manufacturer's instructions. Linked-read sequencing libraries were prepared on the 10× Genomics Chromium platform (10× Genomics). Linked-read libraries have a single sample index, and were pooled to minimize the possibility of barcode swapping between samples from patients who were roommates (see Supplementary Methods). Libraries were sequenced on an Illumina HiSeq 4000 (Illumina).

**Sequence data processing**. TrimGalore version 0.5.0[93] was used to perform quality and adapter trimming with the flags "–clip_R1 15–clip_R2 15–length 60". SeqKit version 0.9.1[94] was used to remove duplicates in short-read data with the command "seqkit rmdup–by-seq". Due to excessive processing time, this step was skipped for linked-read data. Reads were mapped against the GRCh38 assembly of the human genome using BWA version 0.7.17-r1188[95] and only unmapped reads were retained. Quality metrics were verified with FastQC version 0.11.8[96]. Bioinformatics workflows were implemented with Snakemake[97].

**Short-read classification with Kraken2**. We classified all short-read data with a Kraken2[43] database containing all bacteria, viral and fungal genomes in NCBI GenBank assembled to complete genome, chromosome or scaffold quality as of January 2020. Human and mouse reference genomes were also included in the database. A Bracken[44] database was also built with a read length of 150 and k-mer length of 35. Kraken2 version 2.0.8-beta and Bracken version 2.0 were run in paired-end mode with default parameters. Bray–Curtis distances were calculated with the R package vegan[98] version 2.5–7.

**Assembly and binning**. Short-read sequencing samples were assembled using SPAdes version 3.14.0[33] using the '–meta' flag. Linked-read sequencing samples were assembled with Megahit version 1.2.9[34] to generate seed contigs, which were then assembled with the barcode-aware assembler Athena version 1.3[29].

Metagenome-assembled genomes (MAGs) were binned with Metabat2 version 2.15[35], Maxbin version 2.2.7[36] and CONCOCT version 1.1.0[37] and aggregated using DASTool version 1.1.1[38]. MAG completeness and contamination was evaluated using CheckM version 1.0.13[39] and MAG quality was evaluated by the standards set in[40]. All assembled contigs were classified with Kraken2 using the database described above. To generate bin identifications, contig classifications were pooled such that contigs making up at least two-thirds the length of the bin were classified as a particular species. If a classification could not be assigned at the species level, the process was repeated at the genus level, and so on.

**Genome de-replication and SNP profiling**. MAGs were filtered to have minimum completeness 50% and maximum contamination 15% as measured by CheckM, then were de-replicated with dRep version 2.6.2[42] with default parameters except the primary clustering threshold set to 0.95. In further steps, a single de-replicated genome will be referred to as a cluster. Reads from all samples were mapped against the de-replicated set of genomes with BWA. Clusters that had >1× average coverage in at least two samples were retained for further analysis. Individual bam files were extracted for each sample–cluster pair with at least 1× coverage. Bam files were randomly subsetted to a maximum of 2 million reads for computational efficiency. Alignments were profiled and then compared across samples with inStrain version 1.3.11[48] using default parameters.

**Building phylogenetic trees**. MAG average nucleotide identity (ANI) trees (Figs. 2 and 3) were created using the pairwise alignment values from dRep, which uses the MUMmer program[99]. MAGs were filtered to be at least 75% the length of the mean length of reference genomes used in the tree. Reference genomes were selected by searching literature for collections of well-described isolates with genomes available. References that were not relevant and clustered in isolated sections of the tree were removed. Pairwise ANI values were transformed into a distance matrix and clustered using the 'hclust' function with the 'average' method in R version 4.0.3[100]. Heatmaps were created using pairwise popANI values from inStrain, transformed into a distance matrix, and hierarchically clustered using the 'ward.D2' method.

**Determining transmission thresholds**. To determine the ANI threshold to call a comparison a "putative transmission event" we evaluated the distributions of ANI values for within- and between-patient comparisons for different species (Fig. S3). We often detected zero population SNPs in time course samples from the same patient, including *E. faecium* in a pair of samples collected from the same patient 323 days apart. Meanwhile, between-patient comparisons typically had lower ANI values. To verify that transmission events would also result in population ANI values near 100%, we examined external datasets where transmission of bacteria in the microbiome is known to occur as a "positive control". We gathered sequencing data from stool samples of matched mother-infant pairs[18] and fecal microbiota transplantation donors and recipients[23] and processed them with the same methods. In these datasets, we regularly observed genomes with 100% popANI between matched individuals, and did not find cases of 100% popANI between unmatched individuals (Fig. S4). In the ideal cases, we expect transmission of bacteria between the microbiomes of HCT patients to result in genomes with 100% popANI. However, the measured genomes may not reach this level of identity, due to mutations since the transmission event, sequencing errors, or other factors. Therefore, we set the transmission threshold at 99.999% popANI, equivalent to 30 population SNPs in a 3 megabase (Mb) genome. Although this threshold is stringent, we recognize that it may allow for false positives where two closely related strains exist in different patients solely by chance.

Despite our efforts to minimize the impact of barcode swapping on detecting transmission (see Supplementary Methods), we still identified many comparisons with >99.999% popANI that we believe to be false positives. These were filtered out based on the following criteria: short read samples from different patients that were sequenced on the same lane and shared one index sequence or linked read samples that were sequenced on the same lane and reads mapping to the organism shared >40% of barcodes. We also removed pairs that could be affected by "secondary" swapping, where the two samples were not directly affected, but an interaction between other samples from the two patients could cause false positives. In total, we removed 31 comparisons from the final table with >99.999% popANI.

**Pairwise MAG comparison**. MAGs were aligned with the mummer program using default settings[99] and filtered for 1-1 alignments. Dotplots were visualized with the "Dot" program[101] filtering for non-repetitive alignments ≥1 kb.

**Antibiotic resistance gene detection**. Antibiotic resistance genes (ARGs) were profiled in contigs from all samples using Resistance Gene Identifier version 5.1.1 (RGI) and the Comprehensive Antibiotic Resistance Database (CARD)[102] with default parameters. Genes were counted if they met the "strict" or "perfect" threshold from RGI. ARGs were annotated both if they occurred on a contig in the MAG of interest, or anywhere in the metagenomic assembly.

**Isolation, culture and identification of VRE organisms**. Stool samples from patients 11342 and 11349 were resuspended in glycerol and streaked on SpectraVRE (R01830, Thermo Fischer Scientific) plates and incubated at 35 °C overnight. The following day, four colonies from each of the plates that displayed growth were picked and streaked out on a separate quadrant of fresh SpectraVRE plates. These Round 1 (R1) plates were incubated at 35 °C overnight and checked for growth the following day. For each R1 plate, a colony was picked from each quadrant and streaked out on a new quadrant of a fresh SpectraVRE plate. These Round 2 (R2) plates were incubated at 35 °C overnight and checked for growth the following day. For each R2 plate, a colony from each quadrant was subjected to MALDI-TOF bacterial species identification analysis on a Bruker Biotyper (Burker) per manufacturer instructions

**Reporting summary**. Further information on research design is available in the Nature Research Reporting Summary linked to this article.

## Data availability

Raw sequence data for this manuscript, when not previously published, have been uploaded to NCBI SRA under project number PRJNA707487. MAGs generated in this study are available from Zenodo in a tar.gz archive (8.4 Gb) under record number 5768708. Additional information on patient clinical metadata, sequencing datasets generated, statistics of MAGs generated, kraken2 classification results, ANI and inStrain results, days of overlap between patients and antibiotic prescription are available as Supplementary Data files. The kraken2 classification database was built from genomes contained in NCBI Genbank [https://www.ncbi.nlm.nih.gov/genbank/]. The Comprehensive Antibiotic Resistance Database is available online [https://card.mcmaster.ca/].

## Code availability

Workflows for data processing, Kraken2 Classification, MAG binning, and inStrain comparisons can be found at https://github.com/bhattlab/bhattlab_workflows[103]. A workflow to process kraken2 results into taxonomic matrices is available at https://github.com/bhattlab/kraken2_classification[104].

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

## Acknowledgements

We would like to acknowledge the contributions of several members of the Bhatt lab for their assistance in the experimental design, laboratory work and manuscript preparation, including C. Severyn, D. Maghini, S. Vance, A. Natarajan, P. West, A. Behr, M. Chakraborty, and S. Zlitni. We would also like to thank B. Wu at Stanford Health Care for providing information on the geographic layout of the HCT ward, K. Chen for providing information relevant to restrictions on patients consuming probiotics, A. Sidow, and Z. Weng for their instruction and assistance in preparing linked-read sequencing libraries, K. Siranosian for edits to the manuscript, and S. Madison for providing information on HCT patient isolation practices. We would like to thank the Stanford Clinical Microbiology Lab for assistance in identifying isolated and cultured bacterial strains. This work was supported by NIH P01 CA049605 (PI: R. Negrin; Cell Therapy for Leukemia and Lymphoma), and NIH R01 AI143757, CDC BAA 75D30118C02921, Sloan Foundation Fellowship, V Scholar award, and a Damon Runyon Clinical Investigator Award to A.S.B. B.A.S. is supported by a NIST JIMB training grant. Computing costs were supported via a NIH S10 Shared Instrumentation Grant 1S10OD02014101. This work utilized computing resources provided by the Stanford Genetics Bioinformatics Service Center. Finally, we would like to acknowledge the patients and nurses on the Blood and Marrow Transplantation service for their enthusiastic participation in this project.

## Author contributions

B.A.S., A.S.B., H.T., N.B., T.A. designed the study. B.A.S. performed laboratory and computational work, conducted the analysis and wrote the manuscript. E.F.B. performed laboratory work and contributed to writing the manuscript. T.A. performed laboratory work. A.R.R. and T.A. Designed the stool sample collection effort. N.B. assisted with analysis of BSI data and bacterial culture identification. H.T. assisted with computational analysis. A.S.B. oversaw the project and contributed to writing of the manuscript. All authors contributed to editing the manuscript.

## Competing interests

The authors declare no competing interests.
