## [Peer Review File · Nature Communications]

REVIEWER COMMENTS

Reviewer #1 (Remarks to the Author):

We thank the authors for revising the manuscript and adding additional information in response to our comments. This is the largest shotgun metagenomic sequencing dataset from allo-HCT patients so far indicating that patient-patient transmissions of gut bacterial pathogens and commensal are not frequent despite heavy antibiotic use in these patients. As the authors stated themselves, this is a descriptive analysis of existing data and larger and prospective studies including more variables are needed. Methodology meets the expected standards and provides enough detail.

Reviewer #2 (Remarks to the Author):

I mainly just have a few nitpicks left but my main concerns have all been answered.

Previous studies of the microbiome in HCT patients have often used 16S rRNA sequencing^{10,26– 87 28}, which is sufficient for taxonomic classification but cannot differentiate specific strains in a mixed 88 community.

This depends somewhat on the species and V16 region used. With V4 sequencing you can for example differentiate various *F. prausnitzii* strains from one another however some strains, even of *F. prausnitzii*, might still have an identical V4 region and will hence be lumped together. Hence, 16S sequencing lacks the resolution to provide sufficient certainty whether particular signals are identical. It can often provide enough resolution and certainty that strains are not identical.

230 patients have newly detectable *E. faecium* than *E. coli*,

Species names should be in italics.

Length of hospital overlap was significantly associated with having a more similar microbiome strain 275 (linear regression on log-scaled popANI values, $p=0.0017$), while time of roommate overlap was not 276 significantly related (Figure 2i,j).

Does this not suggest that patients, if they do pick up something, pick it up from the hospital? Your cases seem to suggest that this may indeed be the more common route in the case of *Enterococcus*. Also, where are Figures 2i and 2j? It only goes up to h. I think you meant g and h.

This sequence type is 315 believed to colonize the intestinal tract even in healthy individuals without antibiotic exposure⁵⁷, and there 316 are reports of this pathogen causing urinary tract infections in multiple individuals within a household⁵⁸.

Remove “,” or start new sentence?

333 *E. coli*

Italics.

This result suggests that all *E. coli* strains observed 398 are patient-specific, and argues that there are not common strains circulating in the hospital environment 399 or passing between patients.

Remove “,”.

Alternatively, patient-patient transmission or acquisition of common 400 environmental strains is either notably rare or rapid genetic drift after a patient acquires a new strain is 401 reducing popANI levels below the threshold.

Does rapid genetic drift not seem unlikely as you'd then also expect to see it in mother-infant pairs? Or is rapid drift thus not host dependent but dependent on the existing microbiome?

Figures 5 & 6: Genus names are not in italics. Typically, family level and below should be in italics (in my experience). I don't mind myself but the journal might.

Reviewer #3 (Remarks to the Author):

The authors have done a nice job responding to all of the points raised previously. I believe the manuscript is now more balanced and does a better job of being self-aware and exploring the possibility of transmission through a microbiome lens. I do not have any major issues to raise; only the following minor points.

The use of the word "overlap" is a little confusing and inconsistent. In Fig 5 and other locations, the terms, "hospital overlap" and "roommate overlap" are used. I understand that "roommate overlap" is patients that share a room at the same time, and "hospital overlap" is patients in the hospital at the same time. However I was not sure if "roommate" is a subset of "hospital", or if they are mutually exclusive, and whether some minimum amount of temporal overlap was required. Based on the context, my guess is that slightly different definitions were used at different times. Also, Table 2 refers to "number of patients overlapped", it was unclear if this refers to hospital overlap. Would adjust the wording to make the vocabulary more consistent.

Line 80 should be "Clostridioides"

Line 125 should be "multi-drug resistant"

Figure 2: change "bactrim" to "trimethoprim-sulfamethoxazole"

Reviewer #1 (Remarks to the Author):

We thank the authors for revising the manuscript and adding additional information in response to our comments. This is the largest shotgun metagenomic sequencing dataset from allo-HCT patients so far indicating that patient-patient transmissions of gut bacterial pathogens and commensal are not frequent despite heavy antibiotic use in these patients. As the authors stated themselves, this is a descriptive analysis of existing data and larger and prospective studies including more variables are needed. Methodology meets the expected standards and provides enough detail.

We thank the reviewer for recognizing the improvements in the revised manuscript and the strengths of this large dataset.

Reviewer #2 (Remarks to the Author):

I mainly just have a few nitpicks left but my main concerns have all been answered.

We thank the reviewer for recognizing the strengths in the revised manuscript and have responded to the remaining concerns and suggestions below.

Previous studies of the microbiome in HCT patients have often used 16S rRNA sequencing 10,26– 87 28, which is sufficient for taxonomic classification but cannot differentiate specific strains in a mixed 88 community.

- This depends somewhat on the species and V16 region used. With V4 sequencing you can for example differentiate various *F. prausnitzii* strains from one another however some strains, even of *F. prausnitzii*, might still have an identical V4 region and will hence be lumped together. Hence, 16S sequencing lacks the resolution to provide sufficient certainty whether particular signals are identical. It can often provide enough resolution and certainty that strains are not identical.

We agree that our previous wording lacked specificity, and have changed the sentence in the manuscript to “Previous studies of the microbiome in HCT patients have often used 16S rRNA sequencing, which is sufficient for taxonomic classification but may not be sensitive enough to differentiate strains with similar genomes.”

Length of hospital overlap was significantly associated with having a more similar microbiome strain 275 (linear regression on log-scaled popANI values, $p=0.0017$), while time of roommate overlap was not 276 significantly related (Figure 2i,j).

- Does this not suggest that patients, if they do pick up something, pick it up from the hospital? Your cases seem to suggest that this may indeed be the more common route in the case of *Enterococcus*. Also, where are Figures 2i and 2j? It only goes up to h. I think you meant g and h.

We agree that this correlation is interesting - our hypothesis and reasoning follows.

1. We anticipate the reason we did not see a similar correlation in the roommate overlap data is because we are underpowered with the limited number of patients who were roommates in our dataset.
2. Patients do not regularly converge on a “hospital associated microbiome,” indicated by the fact that taxonomic similarity between patient microbiomes does not converge upon hospital or roommate overlap (Figure 2e,f).
3. Based on our threshold of 99.999% popANI defining a shared strain event, patients are also not frequently acquiring common hospital-associated strains, or transmitting strains between each other, as the majority of the data points are below that threshold.
4. One hypothesis for the observed correlation is that various similar, but not identical, strains are present in the hospital and within individuals, and a subset of these strains are passed from one person to another, or from the hospital environment to a person. We have added a clarifying point in the text about Figure 2h (line 290).

Thank you for noting the labeling error; we have fixed the figure labeling as suggested.

Alternatively, patient-patient transmission or acquisition of common environmental strains is either notably rare or rapid genetic drift after a patient acquires a new strain is reducing popANI levels below the threshold.

- Does rapid genetic drift not seem unlikely as you’d then also expect to see it in mother-infant pairs? Or is rapid drift thus not host dependent but dependent on the existing microbiome?

Thank you for the insightful question. We have edited the references to “genetic drift” upon editorial request, now framing the statement as “rapid evolution.” We agree this explanation is unlikely to have a large effect over the short timescales we study here. However, as an alternative explanation to finding transmitted strains, we feel it deserves discussion. There is literature precedent for rapid evolution in the gut upon acquisition of new microbes (refs 82-85). In contrast to mother-infant pairs, these are patients who are undergoing treatment with DNA damaging agents and broad broad-spectrum antibiotics, which may have unknown effects on the rate of microbial evolution.

230 patients have newly detectable *E. faecium* than *E. coli*,

- Species names should be in italics.

This sequence type is 315 believed to colonize the intestinal tract even in healthy individuals without antibiotic exposure⁵⁷, and there 316 are reports of this pathogen causing urinary tract infections in multiple individuals within a household⁵⁸.

- Remove “,” or start new sentence?

333 *E. coli*

- Italics.

This result suggests that all *E. coli* strains observed 398 are patient-specific, and argues that there are not common strains circulating in the hospital environment 399 or passing between patients.

- Remove ”,”.

Figures 5 & 6: Genus names are not in italics. Typically, family level and below should be in italics (in my experience). I don’t mind myself but the journal might.

We thank the reviewer for catching these five points and have fixed them in the revised manuscript.

Reviewer #3 (Remarks to the Author):

The authors have done a nice job responding to all of the points raised previously. I believe the manuscript is now more balanced and does a better job of being self-aware and exploring the possibility of transmission through a microbiome lens. I do not have any major issues to raise; only the following minor points.

We thank the reviewer for recognizing the strengths in the revised manuscript and have responded to the remaining concerns and suggestions below.

The use of the word “overlap” is a little confusing and inconsistent. In Fig 5 and other locations, the terms, “hospital overlap” and “roommate overlap” are used. I understand that “roommate overlap” is patients that share a room at the same time, and “hospital overlap” is patients in the hospital at the same time. However I was not sure if “roommate” is a subset of “hospital”, or if they are mutually exclusive, and whether some minimum amount of temporal overlap was required. Based on the context, my guess is that slightly different definitions were used at different times. Also, Table 2 refers to “number of patients overlapped”, it was unclear if this refers to hospital overlap. Would adjust the wording to make the vocabulary more consistent.

We appreciate the opportunity to make our manuscript more clear. We have revised the text to clarify the use of the word “overlap” and have added an additional explanation at line 137 in the results. To align usage throughout the manuscript, a minimum overlap time of 24 hours has been applied to both roommate and hospital overlap calculations. This changed figures 2e and 2g slightly (which previously had a one hour minimum), but the conclusion about the lack of correlation remains the same. The reviewer is correct in their interpretations of “roommate overlap” and “hospital overlap.” Roommate overlap is a subset of hospital overlap (as patients must be in the hospital at the same time to be roommates).

Line 80 should be “Clostridioides”

Line 125 should be “multi-drug resistant”

Figure 2: change “bactrim” to “trimethoprim-sulfamethoxazole”

We thank the reviewer for catching these three points and have fixed them in the revised manuscript.